# Hard Attention Control By Mutual Information Maximization

## Abstract

Biological agents have adopted the principle of attention to limit the rate of incoming information from the environment. One question that arises is if an artificial agent has access to only a limited view of its surroundings, how can it control its attention to effectively solve tasks? We propose an approach for learning how to control a hard attention window by maximizing the mutual information between the environment state and the attention location at each step. The agent employs an internal world model to make predictions about its state and focuses attention towards where the predictions may be wrong. Attention is trained jointly with a dynamic memory architecture that stores partial observations and keeps track of the unobserved state. We demonstrate that our approach is effective in predicting the full state from a sequence of partial observations. We also show that the agent's internal representation of the surroundings, a live mental map, can be used for control in two partially observable reinforcement learning tasks. Videos of the trained agent can be found at https://sites.google.com/view/hard-attention-control.

## 1 Introduction

Reinforcement learning (RL) algorithms have successfully employed neural networks over the past few years, surpassing human level performance in many tasks (Mnih et al., 2015; Silver et al., 2017; Berner et al., 2019; Schulman et al., 2017). But a key difference in the way tasks are performed by humans versus RL algorithms is that humans have the ability to focus on parts of the state at a time, using attention to limit the amount of information gathered at every step. We actively control our attention to build an internal representation of our surroundings over multiple fixations (Fourtassi et al., 2017; Barrouillet et al., 2004; Yarbus, 2013; Itti, 2005). We also use memory and internal world models to predict motions of dynamic objects in the scene when they are not under direct observation (Bosco et al., 2012). By limiting the amount of input information in these two ways, i.e. directing attention only where needed and internally modeling the rest of the environment, we are able to be more efficient in terms of data that needs to be collected from the environment and processed at each time step.

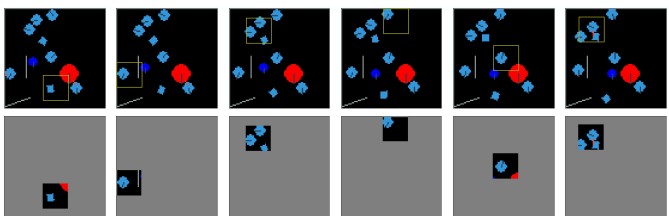

Figure 1: Illustration of the PhysEnv domain (Du & Narasimhan, 2019). Here, the agent (dark blue) needs to navigate to the goal (red) while avoiding enemies (light blue). Typically the full environment states (top) are provided to the RL algorithm to learn. We consider the case where only observations from under a hard attention window (bottom) are available. The attention's location (yellow square) at each time step is controllable by the agent. Using only partial observations, the agent must learn to represent its surroundings and complete its task.

By contrast, modern reinforcement learning methods often operate on the entire state. Observing the entire state simultaneously may be difficult in realistic environments. Consider an embodied agent that must actuate its camera to gather visual information about its surroundings learning how to cross a busy street. At every moment, there are different objects in the environment competing for its attention. The agent needs to learn to look left and right to store locations, heading, and speed of nearby vehicles, and perhaps other pedestrians. It must learn to create a *live* map of its surroundings, frequently checking back on moving objects to update their dynamics. In other words, the agent must learn to selectively move its attention so as to maximize the amount of information it collects from the environment at a time, while internally modeling motions of the other, more predictable parts of the state. Its internal representation, built using successive glimpses, must be sufficient to learn how to complete tasks in this partially observable environment.

We consider the problem of acting in an environment that is only partially observable through a controllable, fixed-size, hard attention window (see figure 1). Only the part of the state that is under the attention window is available to the agent as its observation. The rest must be inferred from previous observations and experience. We assume that the location of the attention window at every time step is under control of the agent and its size compared to the full environment state is known to the agent. We distinguish this task from that of learning *soft* attention (Vaswani et al., 2017), where the full state is attended to, weighted by a vector, and then fed into subsequent layers. Our system must learn to 1) decide where to place the attention in order to gather more information about its surroundings, 2) record the observation made into an internal memory and model the motion within parts of the state that were unobserved, and 3) use this internal representation to learn how to solve its task within the environment.

Our approach for controlling attention uses RL to maximize an information theoretic objective closely related to the notion of surprise or novelty (Schmidhuber, 1991). It is unsupervised in terms of environment rewards, i.e. it can be trained on offline data (states and actions) without knowing the task or related rewards. We discuss this in more detail in section 5. Memory also plays a crucial role in allowing agents to solve tasks in partially observable environments. We pair our attention control mechanism with a memory architecture inspired largely by Du & Narasimhan (2019)'s SpatialNet, but modified to work in partially observable domains. This is described in section 4. Empirically, we show in section 6.1 that our system is able to reconstruct the full state image including dynamic objects at all time steps given *only* partial observations. Further, we show in section 6.2 that the internal representation built by our attention control mechanism and memory architecture is sufficient for the agent to learn to solve tasks in this challenging partially observable environment.

## 2 RELATED WORKS

Using hard attention for image classification or object recognition is well studied in computer vision (Alexe et al., 2012; Butko & Movellan, 2009; Larochelle & Hinton, 2010; Paletta et al., 2005; Zoran et al., 2020; Welleck et al., 2017). Attention allows for processing only the salient or interesting parts of the image (Itti et al., 1998). Similarly, attention control has been applied to tracking objects within a video (Denil et al., 2012; Kamkar et al., 2020; Yu et al., 2020). Surprisingly, not a lot of recent work exists on the topic of hard attention control in reinforcement learning domains, where a sequential decision making task has to be solved by using partial observations from under the attention.

Mnih et al. (2014) proposed a framework for hard attention control in the classification setting and a simple reinforcement learning task. Their approach consists of using environment rewards to train an attention control RL agent. Our approach differs mainly in that we train the attention control using our novel information theoretic objective as reward. Mnih et al. (2014)'s approach leads to a task specific policy for attention control, whereas our approach is unsupervised in terms of the task and can be applied generally to downstream tasks in the environment. Our approach also differs in that we use a memory architecture that is more suited to partially observable tasks with 2D images as input, compared to a RNN used by Mnih et al. (2014).

There has been much prior work on memory and world models for reinforcement learning (Ha & Schmidhuber, 2018; Graves et al., 2016; Hausknecht & Stone, 2015; Khan et al., 2017). The work closest to our own is Du & Narasimhan (2019)'s SpatialNet, which attempts to learn a task-agnostic world model for multi-task settings. Our memory architecture is largely inspired by SpatialNet

and adapted to work in the partially observable setting. We also use their PhysEnv environment to evaluate our approach. Closely related work is Neural Map (Parisotto & Salakhutdinov, 2017), which uses a structured 2D memory map to store information about the environment. Their approach also applies to partially observable RL tasks, but the attention is fixed to the agent. In contrast, we consider the problem of learning to control an attention window that can move independently of the agent location. Recently, Freeman et al. (2019) showed that world models can be learnt by simply limiting the agent's ability to observe the environment. They apply *observational dropout*, where the output of the agent's world model, rather than the environment state, is occasionally provided to the policy. We consider the related scenario where only a part of the true environment state is provided to the agent at each time step and the rest must be modeled using previous observations.

Finally, mutual information has been used to train self-supervised RL agents. This line of work originates in curiosity driven and intrinsically motivated RL (Schmidhuber, 1991; Pathak et al., 2017; Bellemare et al., 2016). Typically, some notion of predictive error or novelty about an aspect of the environment is optimized in lieu of environment rewards. Multiple papers have successfully used different formulations of mutual information to learn how to efficiently explore the environment without extrinsic rewards (Mohamed & Rezende, 2015; Houthooft et al., 2016; Achiam et al., 2018; Gregor et al., 2016; Eysenbach et al., 2018; Sharma et al., 2019). We apply the idea of using mutual information to the problem of curiosity driven attention control.

## 3 PRELIMINARIES

In this work, we are concerned with tasks solved using reinforcement learning (RL). RL is the study of agents acting in an environment to maximize some notion of long term utility. It is almost always formalized using the language of Markov decision problems (MDPs). States, actions, rewards and transitions form the components of an MDP, often represented as the tuple $\langle S, A, R, T \rangle$. Maximizing the expected sum of (discounted) rewards over rollouts in the environment is usually set as the objective of learning. Once a problem is formulated as an MDP, such that the components $\langle S, A, R, T \rangle$ are well defined, one can apply a range of model-free RL algorithms to attempt to solve it (Schulman et al., 2017; Mnih et al., 2016; Wu et al., 2017). A solution to an MDP is typically sought as an optimal policy, $\pi^* : S \rightarrow A$, which is a function that maps every state to an action that maximizes the long term expected rewards. In this section, we will attempt to formalize the components of the partially observable reinforcement learning problem under study.

### 3.1 INTERNAL REPRESENTATION

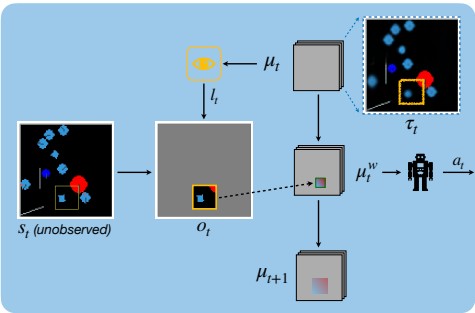

Figure 2: The agent starts with a representation ($\mu_t$) from which it attempts a reconstruction ($\tau_t$) of the likely current full (unobserved) state $s_t$. Then, it picks an attention location ($l_t$) and receives an observation of the state from under the attention ($o_t$). We assume that the size of the attention window is known to the agent. The observation is written into $\mu_t$ to form $\mu_t^w$. The entire map is then stepped through an internal world model to update the dynamical objects, forming $\mu_{t+1}$, the representation for the next step.

First, we give a brief description of how the agent stores observations in an internal representation of its surroundings (figure 2). An map ($\mu$) tracks the full environment state ($s$), which is never directly observed. The map is empty at the start of the episode and gets sequentially written into as observations are made. The agent also creates a reconstruction of the full state ($\tau$) at every time step

based on its map. This will become useful later for training. For now, it suffices that $\mu_t$ is the map *before* an observation is made and $\mu_t^w$ is the one *after* the observation is made. $\mu_{t+1}$ is the map after the dynamics of the system for the next time step are taken into account.

## 3.2 THE TWO AGENTS

We formulate the solutions to controlling the attention at every time step and completing the environment task as two separate reinforcement learning agents. The attention location is controlled by the *glimpse agent* (the eye in figure 2), and actions within the environment are taken by the regular agent. These two agents have their own separate MDPs with their $\langle S, A, R \rangle$ tuples defined below.

The glimpse agent's state at every timestep is $\mu_t$. Its set of actions is all possible attention locations within the full state (Width $\times$ Height actions). Its reward is based on the information theoretic objective discussed in section 5. Thus, the glimpse agent is provided the map before an observation is made and it must decide *where* the observation should be made from in order to optimize its reward.

The environment agent acting in the environment receives as input $\mu_t^w$, i.e. the internal representation after the observation has been recorded. Its actions are the normal set of actions in the environment and its reward is the normal environment reward. We emphasize that neither agent has access to the full state at any time. They must both act based on the internal representation alone. They also cannot make multiple observations from the same environment state. Once an observation is made, an action must be selected that will change the environment state. In the next section, we will describe in detail how $\mu_t^w$, $\mu_{t+1}$ and $\tau_t$ are formed and how the internal map is trained through a sequence of partial observations.

## 4 DYNAMIC MEMORY MAP

Memory plays a critical role in enabling agents to act in partially observable environments. It allows the agent to remember parts of the state that are not under direct observation. Along with memory, the agent must be able to model the dynamics of objects in its environment, as it may not receive observation of them for long periods. Finally, the effect of its own actions must be reflected in its belief about the environment. For this purpose, we design a special recurrent memory unit for visual environments called the Dynamic Memory Map (DMM). DMM is inspired largely by Du & Narasimhan (2019)'s work on SpatialNet, but modified to handle partial observations and work in tandem with the glimpse agent.

### 4.1 MEMORY MODULES

The DMM consists of three major modules: write, step and reconstruct.

**Write.** This module encodes an incoming observation, $o_t$, into the current memory representation, $\mu_t$. The observation is first passed through a series of convolutional operations, $W$, possibly downsampling it for a more efficient representation. Then it is blended with $\mu$ using a series of convolutions, $B$. Finally it is written into the memory but only in the locations where the observation was made, leaving the rest of the memory intact. The write operation can be written as:

$$\mu_t^w = C_t * B(W(o_t), \mu_t) + (1 - C_t) * \mu_t$$

where $C_t$ is a mask which is 1 under the attention window and 0 otherwise.

**Step.** This module is responsible for modeling the dynamics of the environment and updating the memory to track the full state from one time step to the next. Objects in the environment can be static or dynamic and may be affected by the agent's own actions. This module is implemented as a residual network, $S$, similar to SpatialNet (Du & Narasimhan, 2019). We additionally condition $S$ on the agent's own action $a_t$. The entire map is updated at each time step to ensure that objects observed previously are also updated to new likely locations.

$$\mu_{t+1} = \mu_t + S(\mu_t^w, a_t)$$

**Reconstruct.** This module converts the memory representation to a reconstruction of the full state using a series of deconvolutions ($R$).

$$\tau_t = R(\mu_t)$$

A reconstruction can also be made immediately after the observation, i.e. $\tau_t^w = R(\mu_t^w)$. Reconstruction is essential for training the other two modules of DMM as we explain in section 4.2. The reconstruction error, or the discrepancy between the observation and the reconstruction under the attention window can be back-propagated through the write and step layers.

## 4.2 Training loss

The write loss, $L_w$ is incurred under the current attention window immediately after the observation is made into $\mu_t^w$. This ensures that the immediate observation is correctly encoded into the memory and trains the *write* and *reconstruction* modules.

$$L_w = C_t * ||\tau_t^w - o_t||_2.$$

The *step loss*, $L_s$ is incurred after stepping the map and under the attention window in the *next* step.

$$L_s = C_{t+1} * ||\tau_{t+1} - o_{t+1}||_2.$$

This trains the step module, $S$, to accurately model the motion of objects in the entire state, so that a faithful representation of the state can be guessed before the next observation is taken. Once the next observation is made, we can calculate the difference between what the agent's model expected to be at the glimpse location and what actually was there. In figure 2 this is the difference between the images under the yellow rectangle in $\tau_t$ and $o_t$. This quantity is the amount of *surprise* in the dynamics model and we will use this again when training the glimpse agent.

In addition to the reconstruction loss, the absolute value of the output by the *write* and *step* modules is also penalized in order to regularize the contents of the DMM. The total loss for a single step within the rollout is (where $\alpha = \beta = 0.01$ here):

$$L_t = L_w + L_s + \alpha * C_t * |\mu_t^w| + \beta * |S(\mu_t^w, a_t)|.$$

We sum this loss over the entire rollout and jointly minimize it over all steps. The reconstruction error at every location can be back-propagated to the step an observation was recorded there.

## 5 Maximizing Mutual Information to Control Attention

The intuition behind our approach is that attention should be used to gather information from the environment *when and where* it is required. The DMM is constantly modeling its environment and attention should be directed to parts of the world where the model is uncertain. This directly informs the agent about difficult to model parts of the environment, thus reducing uncertainty in the state as a whole. Secondly, by focusing on areas with hard to predict dynamics, more data can be collected from those areas, updating the model for future predictions.

### 5.1 Mutual Information Objective

The idea of reducing uncertainty about the environment can be captured in the language of information theory by using mutual information. Specifically, we propose selecting the location of the attention such that its mutual information with the state at the following step is maximized,

$$\max_{l_t} I(s_{t+1}; l_t). \tag{1}$$

Where $l_t$ is the location of the attention window at time $t$. Eq. 1 can be expanded as

$$\max_{l_t} \mathcal{H}(s_{t+1}) - \mathcal{H}(s_{t+1}|l_t), \tag{2}$$

where $\mathcal{H}$ denotes the entropy. This expansion brings out a very intuitive explanation of the objective. We would like to pick an attention location that maximizes the reduction in entropy (uncertainty) of the environment state after an observation is made.

## 5.2 CURIOSITY DRIVEN ATTENTION

In this section, we show how maximizing the amount of *surprise* can lead to maximizing the mutual information. Let us begin by further expanding eq. 2.

$$
\begin{aligned}
I(s_{t+1}; l_t) &= \mathcal{H}(s_{t+1}) - \mathcal{H}(s_{t+1}|l_t) \\
&= -\sum_{s_{t+1}} p(s_{t+1}) \log p(s_{t+1}) + \sum_{l_t} p(l_t) \sum_{s_{t+1}} p(s_{t+1}|l_t) \log p(s_{t+1}|l_t) \\
&= -\mathop{\mathbb{E}}_{s_{t+1} \sim p(s_{t+1})} [\log p(s_{t+1})] + \sum_{l_t} \sum_{s_{t+1}} p(s_{t+1}, l_t) \log p(s_{t+1}|l_t) \\
&= -\mathop{\mathbb{E}}_{s_{t+1} \sim p(s_{t+1})} [\log p(s_{t+1})] + \mathop{\mathbb{E}}_{s_{t+1}, l_t \sim p(s_{t+1}, l_t)} [\log p(s_{t+1}|l_t)]
\end{aligned}
$$

So, maximizing mutual information between the attention location and the environment state is equivalent to minimizing $\log p(s_{t+1})$ and maximizing $\log p(s_{t+1}|l_t)$ under expectation. The first term is the log-likelihood of the next state prior to selecting $l_t$. Computing it requires marginalization over all possible $l_t$, which is prohibited by our environment as only a single partial observation from the full state can be provided to the agent. The second term, $\log p(s_{t+1}|l_t)$, is computable at a single attention location from a single state and hence we will focus on maximizing this term.

Now, assume the agent's belief over the true environment state at the next step, $s_{t+1}$, is represented by a Gaussian with unit variance around the reconstruction, $\tau_{t+1}$. So, $s_{t+1} \sim \mathcal{N}(\tau_{t+1}, \mathcal{I})$ and

$$
\begin{aligned}
\log p(s_{t+1}|\tau_{t+1}) &\propto \log exp[-(s_{t+1} - \tau_{t+1})^T (s_{t+1} - \tau_{t+1})/2] \\
&= -\sum_i (s_{t+1}^i - \tau_{t+1}^i)^2/2
\end{aligned}
\tag{3}
$$

So, minimizing the squared difference between $s_{t+1}$ and $\tau_{t+1}$ leads to maximizing $\log p(s_{t+1}|\tau_{t+1})$. Maximizing $\log p(s_{t+1}|\tau_{t+1})$ in turn means maximizing $\log p(s_{t+1}|l_t)$, since $\tau_{t+1}$ is constructed using the deterministic functions $W$, $S$, and $R$ once $l_t$ has been picked (see section 4).

In order to minimize the squared difference between $s_{t+1}$ and $\tau_{t+1}$, the agent needs to pick a location $l_t$ that observes the *maximum* error between $s_t$ and its reconstruction $\tau_t$. This is because at step $t$, if the agent observes the location with the highest reconstruction error, i.e. the *least* likelihood $p(s_t|\tau_t)$, it will ensure that at the next time step $t+1$ the agent has the most recent information on that region and can make a good reconstruction of it. Hence, maximizing $\log p(s_{t+1}|l_t)$ is equivalent to picking an $l_t$ that will lead to the highest reconstruction error at step $t$.

$$
\max_{l_t} \log p(s_{t+1}|l_t) \implies \max_{l_t} \sum_{i \in I_t} (s_t^i - \tau_t^i)^2/2
\tag{4}
$$

where $I_t$ forms the set of indices under the window at location $l_t$. Directly optimizing this quantity with respect to $l_t$ is not possible as it can only be computed once an observation has been made and only at a single location. Hence, the glimpse agent must learn to *predict* the location that is most likely to result in high reconstruction error before an observation is made. Our approach is to formulate the glimpse agent as a reinforcement learner with the reconstruction error as its reward.

An interpretation of this objective is that the glimpse agent is surprise seeking, or curiosity driven. It is attending to parts of the state that are novel in the sense that they are difficult for the agent's current model to predict. Another interpretation is that the glimpse agent is acting adversarially to the DMM's model of the environment. At each step the glimpse agent tries to focus on parts of the state that are the most difficult for DMM to reconstruct. By doing so, it is indirectly creating a curriculum of increasingly difficult to model aspects of the environment.

## 5.3 FULL TRAINING OBJECTIVE

Let us look at a different expansion of the objective in eq. 1,

$$
I(s_{t+1}; l_t) = \max_{l_t} \mathcal{H}(l_t) - \mathcal{H}(l_t|s_{t+1}).
\tag{5}
$$

The first term is $\mathcal{H}(l_t)$, the entropy over the attention location, which is controlled by the policy of the glimpse agent. This term can easily be maximized by standard RL algorithms, such as A2C (Mnih et al., 2016; Wu et al., 2017), using weighted entropy maximization.

Combining the first term from equation 5 and second term from equation 2, the final objective that we use to train the glimpse agent is

$$\max_{l_t} \mathcal{H}(l_t) - \mathcal{H}(s_{t+1}|l_t) \equiv \alpha \mathcal{H}(\pi_{glimpse}) + \max_{l_t} \sum_{i \in I_t} (s_t^i - \tau_t^i)^2/2,$$

where $\alpha$ is the entropy weighting set to $0.001$ in our experiments.

Note that this objective is not the mutual information from equation 1. It is possible to design a variational algorithm that maximizes the approximate mutual information based on the expansion in eq. 5 alone (instead of mixing terms from both the expansions). Empirically, however, we found that our objective performs better than the variational approximation to mutual information. See section A.4 for more detail.

## 6 RESULTS

We evaluate our approach in two environments: a gridworld environment (figure 5) and PhysEnv (figure 1). In both environments, the task is to navigate to a goal while avoiding obstacles and enemies. The agent's actions are movement in the four discrete cardinal directions. In the gridworld, the state is a one-hot image of the objects in the environment (agent, walls, enemies and goal). The agent's and enemies' movements are restricted to the grid squares and the dynamics of the enemies are simple: moving in straight lines until an obstacle is hit in which case it reflects. In PhysEnv, observations are provided as RGB images and the motions of the objects are more varied.

The glimpse agent + DMM are first trained on offline data of state and action trajectories (no reward) collected from each environment. More details on data collection are provided in section A.1. We evaluate this part of the training by how accurately the full state is reconstructed by DMM at each step. Once the glimpse agent + DMM have converged, i.e. the reconstruction loss has plateaued, we freeze their parameters and initialize an agent within the environment. The glimpse agent + DMM serve as fixed components within the agent while it is learns to solve a task. The agent is trained using an off-the-shelf RL algorithm PPO (Schulman et al., 2017) and is evaluated by the total environment reward it collects during testing episodes.

We compare our method against baselines inspired from related work. In the *follow* baseline the attention is always focused on the agent itself. This is a formulation of the partial observability as seen in Parisotto & Salakhutdinov (2017). Next, the *environment* baseline mimics the approach by Mnih et al. (2014), where the environment rewards are used to train the glimpse agent. Finally, the *random* baseline moves the attention randomly at each time step. For the environment task, we have an additional baseline, *full*, where the full state is provided to the agent to solve the task. This is an upper limit for how well any method that only receives partial observations can perform.

### 6.1 STATE RECONSTRUCTION

|  | l2 reward (ours) | random | follow | environment |
|---|---|---|---|---|
| Gridworld | **0.00555** | 0.007666 | 0.01941 | 0.00827 |
| PhysEnv | **0.0521** | 0.0614 | 0.1186 | 0.0826 |

Table 1: Per-pixel L2 loss between agent's reconstruction $\tau$ and the ground truth full state $s$. All methods only see partial observations, but must attempt to reconstruct the full state from internal representation. In both environments, our method has the lowest reconstruction error.

We measure the performance of the glimpse agent + DMM by the average reconstruction error to the true full (unobserved) state in unseen testing episodes. 25 consecutive observations from 6 unseen test episodes are passed through each model and error is averaged over reconstructions at each time step. This gauges the ability of our attention mechanism and memory to reconstruct the full state using only partial observations. Table 1 shows that using surprise driven reward (our approach) for

learning attention control leads to a lower reconstruction error than the other baselines. See figures 5 through 9 in the appendix for visualizations of the reconstructions and training error. In particular, figure 7 and 8 show qualitative comparisons between our method with the baselines. Broadly speaking, our method learns to focus on dynamic objects within the environment and preserve them over many steps in the memory, whereas the baselines lose track of or blur objects as seen in their state reconstructions.

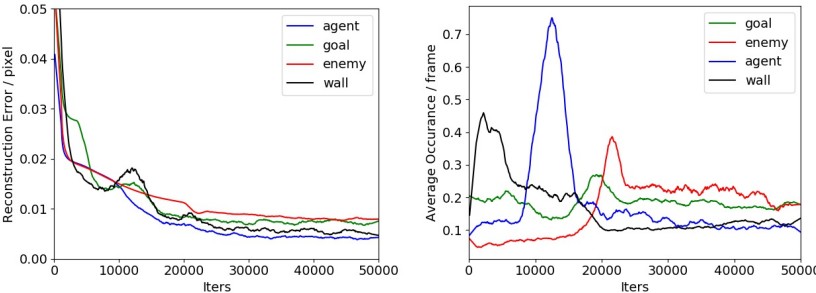

Figure 3: (left) Object-wise breakdown of reconstruction error in gridworld. (right) % of frames in which the object occurs, i.e. the frequency with which the attention is focuses on a particular object.

Figure 3 shows an analysis of what objects are under focus during training in the gridworld environment and their corresponding reconstruction error. The attention forms an interesting curriculum for training the DMM, where it learns to focus on different objects in the environment over time. It starts with walls, then moves to the agent itself, then goal and finally the enemies. Thus, it moves gradually from static to dynamic, harder to model parts of the state, allowing the DMM to learn how to accurately model each object. This very interesting curriculum-like behavior emerges naturally and is not pre-programmed into DMM or the attention agent. It is a consequence of maximizing surprise at every time step. See section A.2 for a more detailed discussion.

## 6.2 REINFORCEMENT LEARNING

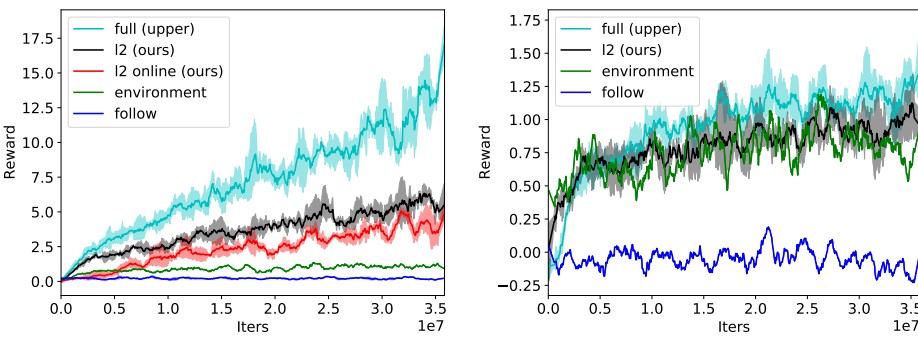

Figure 4: Average testing reward for goal seeking task in PhysEnv (left) and gridworld (right). Comparison between agent with full observations (upper limit) and partial observations trained with surprise (ours), environment rewards and simply following the agent's location.

Figure 4 shows the average episodic reward over five testing episodes during the training of the agent. Our method approaches the performance of the upper bound agent that receives full environment states in the gridworld environment. The *environment* baseline performs similarly to our approach in gridworld, but does poorly in PhysEnv, where our approach performs the best. This may be because the attention policy of the *environment* baseline has high entropy and ends up exploring large parts of the state within an episode. But it does not learn to focus on the most unpredictable parts of the state as our method does. The *follow* baseline performs poorly in both environments, likely because it only focuses on the agent and does not explore the rest of the state.

We also show the performance of our agent (l2 online) in the condition where the DMM and glimpse agent is trained online along with the policy, all components starting from random initialization. This agent did not have a pretraining step and was trained entirely on random exploratory data, therefore obviating the need of a pretraining dataset. The performance is slightly worse than the pretrained agent. This is an encouraging result in a particularly challenging setting of a non-stationary state space as the memory representation and glimpse policy are changing while the agent learns.

## 7 CONCLUSIONS

We have presented an approach of using mutual information to control a hard attention window in environments with dynamic objects. From partial observations under the attention window, our system is able to reconstruct the full state at every time step with the least error. The representation learned using our method enables RL agents to solve tasks within the environment, while the baseline methods are unable to learn useful representations on the same memory architecture. This demonstrates that attention control plays a large role in enabling task completion.

Note that our attention control objective is independent of the task. It is solely concerned with gathering information from the environment where it is most unpredictable. This is similar to curiosity driven RL that seeks novel states or surprising transitions. Hence, attention control learned in this manner is a generic solution that can be applied to many downstream tasks. It may be able to be combined with task-specific attention control by fine-tuning on a task reward. A better foveal glimpse model may also be used instead of the fixed size hard attention used here. We made an assumption that only one glimpse is allowed per environment state. But the rate of collecting glimpses may be variable with respect to environment speed.

Curiosity based approaches tend to suffer from the "noisy TV" problem (Savinov et al., 2018) where the agent tends to get distracted by and focus on random, unpredictable events in the environment. Orthogonal approaches have been developed to counter this common weakness by, for example, taking into account the agent's own actions (Pathak et al., 2017). These approaches can be combined with our method as well. We leave the incorporation of this in attention control for future work.

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

# A APPENDIX

## A.1 DETAILS OF DATA COLLECTION

For gridworld we used 20000 random steps in the environment to create this training set. In PhysEnv we used a previously trained agent to generate 500,000 steps of demonstrations. We experienced that using a random policy in PhysEnv led to quick deaths of the agent with insufficient episode lengths for learning. In both cases, the data are sequences of full states and agent actions from multiple episodes. In PhysEnv, the map size was $21 \times 21$ compared to the full state size of $84 \times 84$ and in gridworld both the map and state were $10 \times 10$. The size of the attention window was fixed to be $3 \times 3$ in the gridworld ($9\%$ of the full state visible in an observation) and $21 \times 21$ in PhysEnv ($6.25\%$ of the state visible).

## A.2 VISUALIZATIONS OF GLIMPSES DURING TRAINING

We will discuss where the attention focuses and how that effects the reconstruction (and underlying belief) as the training progresses for the gridworld environment. Please refer to 3 for this discussion.

Initially, the attention learns to focus on the *wall* objects and the corresponding reconstruction error for the walls decreases rapidly (3.b). This is because the walls are long objects in the environment, present at multiple pixels and side-by-side, providing the largest and most reliable source of error under the attention window initially. The error is provided as reward to the glimpse agent, hence it seeks more walls in the state.

Soon, the error it receives from walls decreases as the agent's DMM learns to persistently record the location of stationary walls over multiple time steps. Then around 10000 iters, the attention switches to focusing on the agent's location itself. This is because the agent's apparent erratic movements provide a larger reward due to their unpredictability than the stationary walls can anymore. This has dual consequences. First, since now the DMM is seeing a lot of examples of the agent's movement, it learns to accurately model the agent based on an initial state and its actions and the reconstruction error corresponding to the agent falls. At its peak, over 70% of glimpses contain the agent within them (figure 3.b). Interestingly, the error corresponding to the wall and goal objects rises as some forgetting of the representation and dynamics of these objects occurs. A little before 20000 iters though, the agent has learnt to represent simultaneously all three objects and it switches attention to the enemies, the only remaining un-modeled objects in the environment. The enemies move predictably, bouncing off walls and other objects, but otherwise in straight lines. The agent does not have direct access to their actions and must infer their motion over multiple time steps using only glimpses.

Figures 5 and 6 visualize the behavior of the glimpse agent during the initial stages of training. Initially, (100 iters), the attention mostly moves around randomly as can be seen by the heatmap being active throughout the state. The reconstruction does not correspond to anything in the real state since the agent has only just begun learning. The glimpse agent here is receiving high reward for pretty much every location in the state. By 2000 iters, the glimpse agent has learnt to mainly focus on the wall once it locates it. This can be seen in the high probabilities of the glimpse agent policy near the wall locations and the attention repeatedly fixating on the wall. The agent is simultaneously striving to learn to represent the walls as they are showing up a lot in its training data, and steadily reducing the reconstruction error (the glimpse agent's reward) coming from walls.

By iter 11000, the focus has shifted to tracking the agent as can be seen in the heatmap. The heatmap shows that the glimpse agent is singularly focused on the agent's movements as the error reward it is getting from the walls has dropped off and the agent's movements are not fully predicatable yet. In the figure for iter 19000, it can be seen that the agent can now track its own location very reliably with an initial position and the actions it is taking, without having to focus the attention on itself. In fact, after step 8 when the agent is discovered, the attention almost never moved back to it and yet the location in the reconstruction perfectly matches that of the unobserved full state. Hence the agent's location based reward source has largely dried up. The heatmap of attention policy shows that the glimpse agent has become interested in the goals and walls again briefly as it learns to represent all three objects together. It can also be seen from the reconstructions here that the agent is unable to accurately track and model the enemies locations at this point.

By iter 21000, the attention has shifted focus to the remaining object in the state that is producing a high reconstruction error, i.e. the enemy. Once the enemy is discovered on step 14, the glimpse agent's policy puts all probability near its location so it can keep observing and model its behavior.

The glimpse agent creates a natural curriculum for training the models within DMM, starting at large stationary objects that do not move, then moving to the agent's followed by the objects with the most complex dynamics. We conjecture this automatically discovered curriculum is the reason our system is able to learn better reconstructions than the baselines.

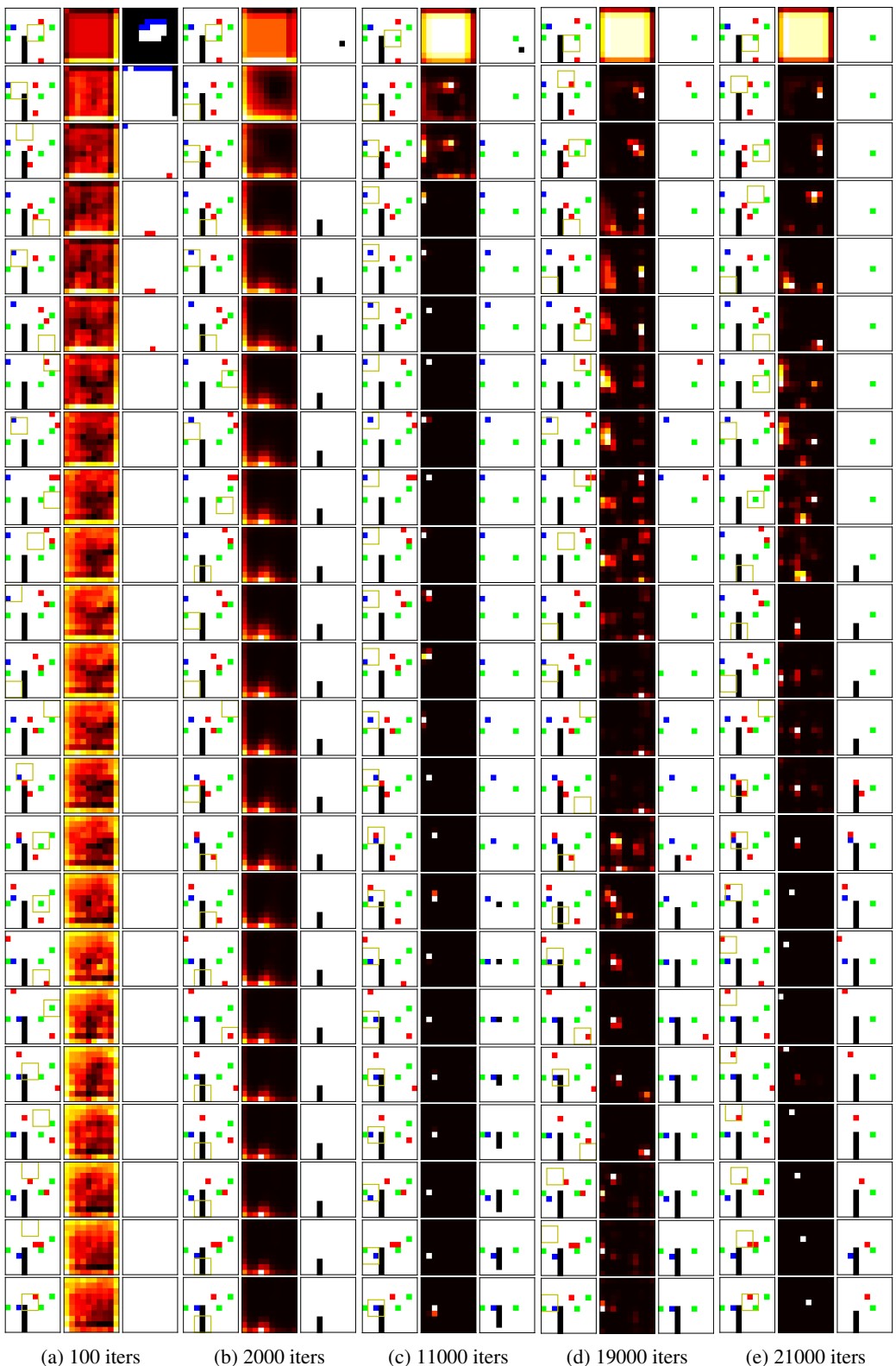

(a) 100 iters  (b) 2000 iters  (c) 11000 iters  (d) 19000 iters  (e) 21000 iters

Figure 5: Visualizations of glimpse behavior in gridworld environment. Each sub-figure is the same testing episode at different stages of training, with the rows corresponding to the time within the episode. In each subfigure, the first column shows the full (unobserved) state. The second column is a heatmap showing the glimpse policy, i.e. the likelihood the glimpse agent will select a location for placing attention. The location that was sampled is indicated in the first column by the yellow square. The final, third, column is the reconstruction $\tau$ based on the agent's current current belief about the environment.

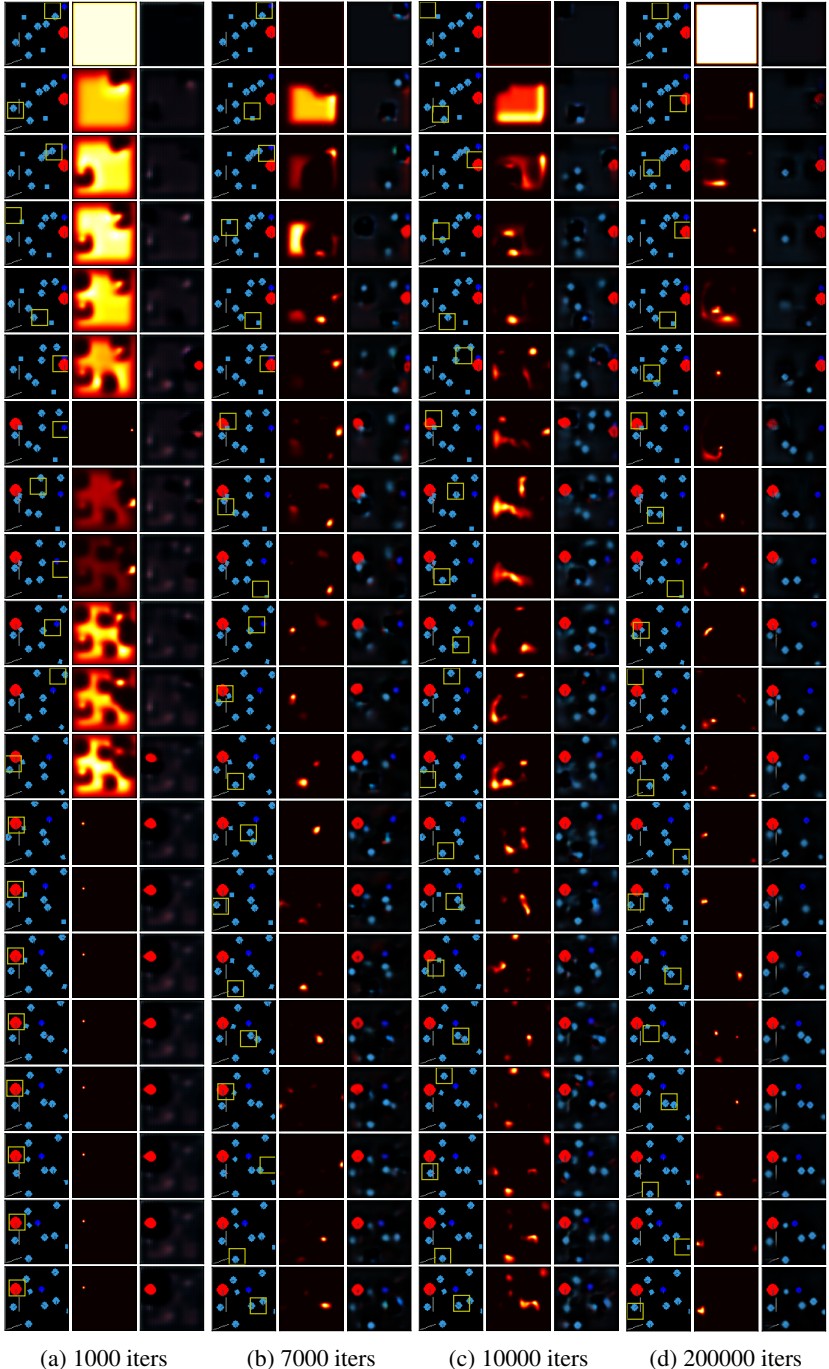

(a) 1000 iters  (b) 7000 iters  (c) 10000 iters  (d) 200000 iters

Figure 6: Visualizations of glimpse behavior in PhysEnv environment.

Next we show qualitative comparisons of the glimpse policy and state reconstruction between our method and the baselines. Each subfigure in figures 7 and 8 shows the same episode of length 20 played out with different glimpse agents. The first column is our method, the second shows a uniform random policy for the glimpse agent, the third shows a glimpse agent trained on environment (task) rewards, and lastly the fourth shows the glimpse following the agent. In both environments, it can be seen that our method learns to focus on dynamic objects within the environment and the resulting state reconstruction shows that these objects are preserved in the memory.

Note that the glimpse policy learned by the environment rewards is significantly more entropic than the one learned by our method, despite having the same entropy weighting hyperparameter. We conjecture that this is because the task reward does not send a clear and consistent learning signal to control the attention window to create better reconstructions, or as seen in our RL results, to facilitate downstream tasks. As such, the reconstructions appear qualitatively similar to the random baseline.

The random baseline quantitatively performs the closest to our approach, but it fails to focus on dynamic objects consistently like over time, and hence fails to track these objects over long periods in the episode. The reconstructions therefore have streaks and blurs where the objects were last spotted and the physics model has not learnt how to update their positions properly.

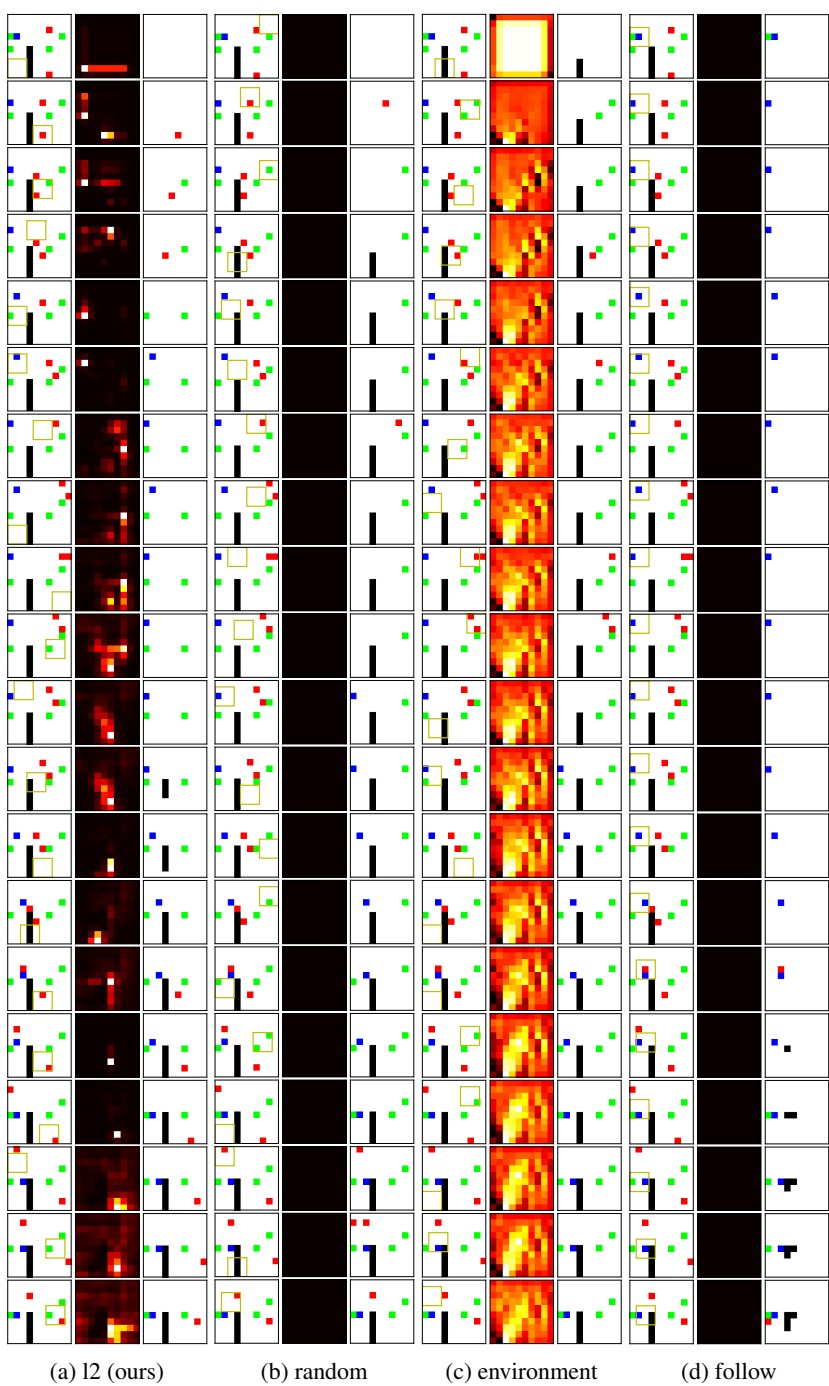

(a) l2 (ours)   (b) random   (c) environment   (d) follow

Figure 7: Glimpse behavior for fully trained models of our method and the baselines in gridworld. Each subfigure is a different baseline, with columns within the subfigure being the full unobserved state, heatmap of attention policy, and reconstruction of full state respectively.

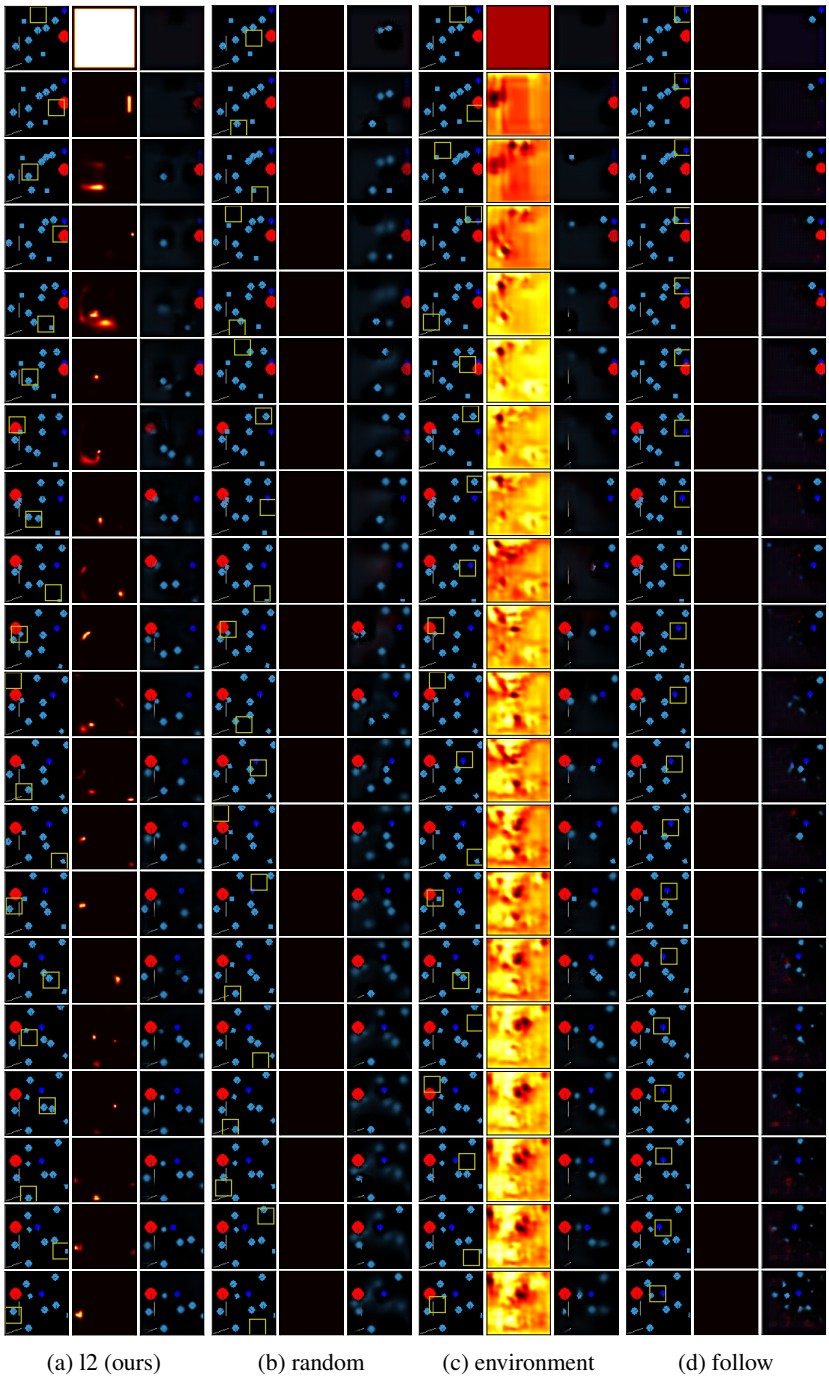

(a) l2 (ours)  (b) random  (c) environment  (d) follow

Figure 8: Glimpse behavior for fully trained models of our method and the baselines in PhysEnv.

### A.3    ERROR CURVES FOR GLIMPSE AGENT + DMM

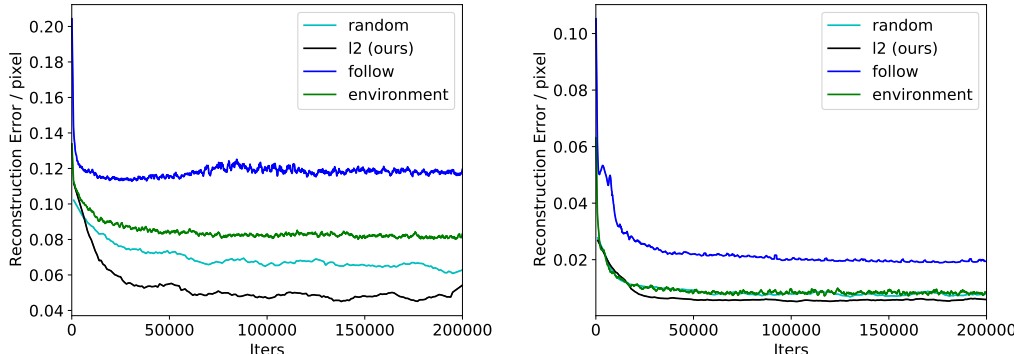

Figure 9: (left) PhysEnv environment. (right) Gridworld environment. Reconstruction error (L2) from the full state during training of the DMM + glimpse agent. All approaches have converged by 200k iters, with our approach having the lowest error, i.e. the most faithful reconstruction of the full unobserved state.

These training curves correspond to the results in table 1. The reconstruction error for each method has stabilized. Our method has the lowest reconstruction error for both environments.

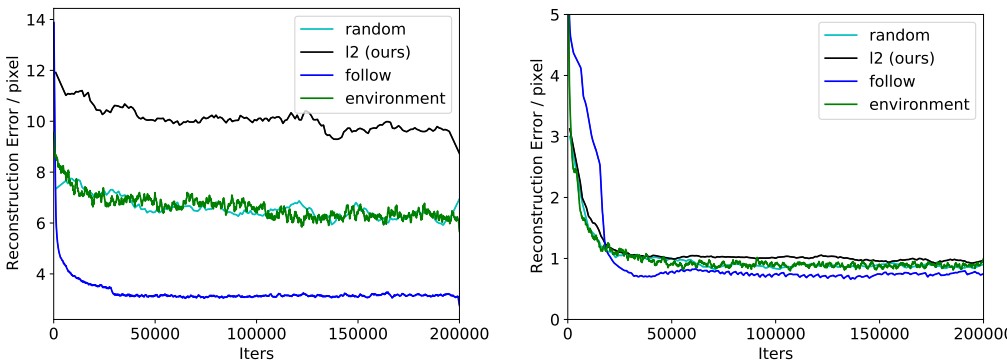

Figure 10: (left) PhysEnv environment. (right) Gridworld environment. Episodic reward for glimpse agent during training (higher is better).

Figure 10 shows the episodic reward for the glimpse agent on six testing episodes as the training progresses. This corresponds to the mutual information objective in equation 4, or the reconstruction eror under the attention window in the *next step*. The modules within DMM are constantly improving to reduce prediction error and hence the reward goes down as the training progresses. But the glimpse agent is simultaneously trying to maximize the objective by searching for areas within the state that still provide high error. Our method maximizes this objective in both environments. This leads to lower overall reconstruction error as the glimpse agent is learning to predict where the highest source of error is going to be at the next step. The environment baseline performs close to random and the follow baseline gathers the least reward in both environments.

## A.4 Variational Approximation to Mutual Information Maximization

The mutual information between attention and environment state as expanded in eq. 5 is

$$I(s_t; l_t) = \mathcal{H}(l_t) - \mathcal{H}(l_t|s_t). \tag{6}$$

In our algorithm, we combine the first term in this expansion with another term in eq. 2. This works well empirically. Theoretically, however, it is computationally possible to maximize both terms in eq. 6, hence maximizing the actual mutual information. However, as we shall show here, this does not work well practically.

The first term can be maximized by using maximum entropy RL algorithms for the attention policy as discussed in the main text. Let us consider the second term.

$$
\begin{aligned}
-\mathcal{H}(l_t|s_t) &= -\sum_{s_t} p(s_t)\mathcal{H}(l_t|s_t) \\
&= \sum_{s_t} p(s_t) \sum_{l_t} p(l_t|s_t) \log p(l_t|s_t) \\
&= \mathbb{E}_{s_t, l_t \sim p(s_t, l_t)}[\log p(l_t|s_t)]
\end{aligned}
$$

While $\log p(l_t|s_t)$ is not directly known, we can construct a variational function, $q(l_t|s_t)$, to approximate it.

Since $KL(p(l_t|s_t)||q(l_t|s_t)) \geq 0$, we have,

$$
\begin{aligned}
\sum_{l_t} p(l_t|s_t) \log p(l_t|s_t) &\geq \sum_{l_t} p(l_t|s_t) \log q(l_t|s_t) \\
\mathbb{E}_{s_t, l_t \sim p(s_t, l_t)}[\log p(l_t|s_t)] &\geq \mathbb{E}_{s_t, l_t \sim p(s_t, l_t)}[\log q(l_t|s_t)]
\end{aligned}
$$

Plugging this into eq. 6,

$$I(s_t; l_t) \geq \mathcal{H}(l_t) + \mathbb{E}_{s_t, l_t \sim p(s_t, l_t)}[\log q(l_t|s_t)]$$

Therefore, we can maximize the mutual information by maximizing this lower bound w.r.t. the policy parameters of the glimpse agent. This can be achieved by providing $\log q(l_t|s_t)$ as a reward to the glimpse agent and using RL algorithms to maximize its long term value along with the policy entropy. As such, maximizing the mutual information.

We can additionally condition $q$ on the internal state and action of the agent at previous time step. Since we do not have access to the full environment state, $s_t$, we substitute it with the agent's internal state. The final variational function looks like $q(l_t|\mu_t, \mu_{t-1}, a_{t-1})$. $q$ must be trained to match the true posterior $p(l_t|s_t)$ to provide accurate rewards to the glimpse agent.

The question now is how to learn $q$? In this work, it is a neural network trained on samples. This is done by minimizing the KL divergence,

$$\min \mathbb{E}_{s_t, l_t \sim p(s_t, l_t)}[KL(p(l_t|s_t)||q(l_t|\mu_t, \mu_{t-1}, a_{t-1}))]$$

on samples from the environment and the glimpse agent's policy. This will ensure $q$ is likely to assign high likelihood to actual locations conditioned on its inputs.

Intuitively, $q$ will be high at locations of attention that are easy to identify by looking at the previous state, the resulting state, and the agent action. Otherwise, its output will be flat and hard to predict. Since $q$ is provided to the glimpse agent as reward, it will be rewarded for selecting easily identifiable locations and prefer such a policy. These can be locations that produce a high amount of localized surprising information in $\mu_t$ as compared to $\mu_{t-1}$, as unexpected appearance of goal or enemies somewhere will signal that the attention was just moved there, hence producing a high likelihood in $q$. Conversely, low reward locations will be ones that are hard to identify by $q$ as they do not impact any specific part of $\mu_t$, leaving it largely the same as $\mu_{t-1}$, or observe a location that was already well-modeled by the agent and hence do not leave an identifiable *mark* on $\mu_t$ that is a tell-tale sign for $q$.

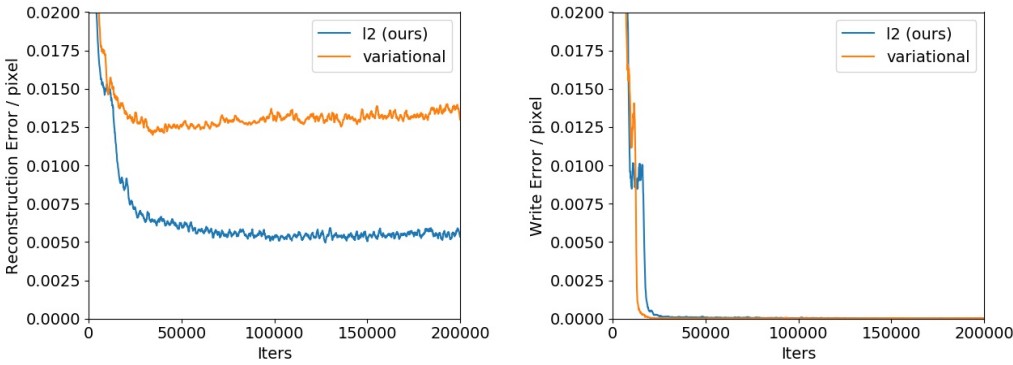

(a) Overall state reconstruction error per pixel.    (b) Write error of observation under attention window.

Figure 11: State reconstruction and observation reconstruction errors in gridworld environment. Observation reconstruction error falls to almost zero for both methods while full state reconstruction remains high for variational approach.

The above method does not work well in practice. Figure 11a shows the reconstruction error of the unobserved state with our proposed method and the variational approach. Figure 11b shows the observation reconstruction error, which is the error in reproducing just the observation under the attention immediately after it is made. It can be seen that the latter quantity goes to near zero for both methods, meaning that immediate observations are being faithfully recorded into memory and reconstructed. Yet, the overall state reconstruction error remains high for the variational approach, which means that the agent is not able to retain observations over time steps as the attention moves around the state and the environment model is not learning to predict the evolution of the state over time without direct observation.

A reason why the variational approach fails in practice could be that we do not have direct access to $p(l_t|s_t)$ in order to train $q$. Instead we must rely on the agent's internal estimate of the environment state, $\mu_t$, which itself is being trained alongside the glimpse agent. This results in noisy rewards to the glimpse agent as the $q$ function is not able to accurately predict where the attention is. As we have seen in section A.2, attention control creates a curriculum for training the agent's state estimate. Hence this forms a loop, where attention control is dependent on agent's state estimate which is in return dependent on the curriculum generated by the attention.

Extensive hyperparameter tuning may also be required for this approach to work as the glimpse agent can easily default to chosing the same attention location at each time step as it ensures the highest $q(l_t|s_t)$.

