# OpenReview forum: "Hard Attention Control By Mutual Information Maximization"
_ICLR.cc/2021/Conference — Reject_

### Official Review · AnonReviewer1 · 2020-10-27
**Reinforcement with Hard-attention is interesting.**

**Rating:** 6
**Confidence:** 2

**Review:**

Summary
1. This paper proposes a new way to solve tasks under the limited view of the surroundings.
2. To solve the tasks, three components, 1) decide where to place the attention, 2) record the observation and make internal memory, 3) solve the task with the memory, are necessary
3. This paper utilizes hard attention with mutual information technique to solve the problems, and mutual information objective reduces the uncertainty about the environment.
4. The proposed system of this paper is able to reconstruct the full state at every time step.
5. As a result, the l2 reward shows the lowest state reconstruction and the highest reward except full (upper).

Strengths
1. The problem definitions and the proposed models are interesting and clear.
2. This paper investigates the unexplored research area, hard attention in reinforcement learning domains
3. This paper provides the demo videos and implementation codes.

Weaknesses
1. This paper has empirical contributions, but the technical contributions are not clarified.
2. The results are not enough to support the author's claims.
The interesting qualitative results are provided in the appendix and videos.
However, there is no comparison between baselines and this paper qualitatively.

Questions and Additional Feedback
1. The intuition of total loss in section 4.2 is not well described. It is hard to follow that the necessity of regularization term and alpha (in total loss and entropy weighting), beta.
2. How much alpha (in total loss and entropy weighting) and beta influence the performance? (sensitive analysis)
3. L_t in 4.2 is a total loss, but L_t in 5.2 denotes the indices. I suggest that this paper should be modified carefully.
4. Table 1 denotes the reconstruction loss, but the difference between random and l2 reward seems doesn't large.
Is there another experiment or settings that l2 reward solves well, but random fails to solve?
5. Figure 4 shows the test performance, and the gap between full and l2 is very large.
What is the reason that l2 performs worse than full on the PhysEnv, while full and l2 shows similar performance on the gridworld?
6. Could you summarize your technical contributions instead of empirical contributions?

Typos
Emperically -> Empirically


After Rebuttal
Thank you for your detailed response, and I will keep my positive score.

---

> ### Author Response · Authors · 2020-11-24
> **Response R1**
>
> Thank you for your valuable feedback and encouraging words! Here are responses to specific questions:
>
> -- “However, there is no comparison between baselines and this paper qualitatively.”
> We have added qualitative comparisons between our method and the baselines in the appendix (figures 7 and 8).
>
> -- “The intuition of total loss in section 4.2 is not well described.”
> In section 4.2, alpha and beta are used to control the impact of regularization on the total loss. Regularization is added to stabilize the training of the memory representation through sparsity. It is a standard technique for stabilization used in autoencoder style models. (Higgins et al., 2016)
>
> Higgins, Irina, et al. "beta-vae: Learning basic visual concepts with a constrained variational framework." (2016).
>
> -- “Table 1 denotes the reconstruction loss, but the difference between random and l2 reward seems doesn't large.”
> While the difference may not appear large (the error is between normalized pixel values between [0,1] and is calculated per pixel), the qualitative results in figures 7 and 8 make it clear that our method outperforms the “random” baseline in representing the full state in memory. Please refer to the discussion in section A.2 on those figures as well (page 16).
>
> -- “Figure 4 shows the test performance,...”
> PhysEnv is overall a harder task with partial observability as the attention window is observing a smaller fraction of the entire state (9% in gridworld vs. 6.25% in PhysEnv) and the environment dynamics are much harder to model. In PhysEnv, the objects do not have to stay along the grid lines and are more in number as well.
>
> -- “Could you summarize your technical contributions instead of empirical contributions?”
> We provide a mutual information objective for controlling hard attention in RL settings and show that it is equivalent to maximizing the reconstruction error in the following step. We outline a system where an RL agent learns to maximize this objective through samples and a memory architecture that keeps track of the full state simultaneously.
>
> Thank you for the minor suggestions.

---

### Official Review · AnonReviewer3 · 2020-10-28
**Significance of the contribution is unclear.**

**Rating:** 4
**Confidence:** 4

**Review:**

This work restricts agent observations to a small viewport within the full image representing the environment’s state, and trains one agent to control the location of this viewport so as to reduce the reconstruction error of an internal model of the full image state. The proposed method produces lower reconstruction error than baselines tested, and the reconstructed image is used to train another agent to obtain reward from the environment.

Pros
- The work is carefully motivated and focused on the question of how to successfully control hard attention.
- The solution (observing the location with the highest predicted reconstruction error) is intuitively appealing.

Cons
- The two environments considered are small and simple.
- The memory is structured to match the environment’s actual image state, and the memory write operation is automatically masked to the viewport’s corresponding location within the memory. This contrived arrangement injects a strong inductive bias that seems unlikely to generalize to environments that don’t follow these strict assumptions.
- The automatic masking reduces the challenge of state reconstruction.
- The reasoning related to mutual information seems circular. It begins with the intuition of using attention to gather information where it is needed, and ends with the minimization of reconstruction error, which is essentially the starting point. The mathematical excursion provides little insight beyond the original intuition.

Suggestions
- An intrinsic reward based on image reconstruction could drive the agent’s exploration during an initial pretraining phase, where no external reward is provided. If the agent learns enough about its environment during this first (model-building) phase, it may learn more quickly than the full-observation oracle during a second phase of training, where the external reward is provided. Such a demonstration of benefit from restricted observations would be an important contribution, if it held for more complex environments, and if the results included systematic hyperparameter tuning and multiple runs using different random seeds.

---

> ### Author Response · Authors · 2020-11-24
> **Response R3**
>
> Thank you for your feedback and encouraging words. In response to your particular questions:
>
> -- “The memory is structured to match the environment’s actual image state,...”
> -- “The automatic masking reduces the challenge of state reconstruction.”
> Yes, the memory is on-purpose designed for partially observable environments like this. It is a modification of the SpatialNet framework which increases efficiency when only a hard attention window is observed. The assumptions of the size of the attention window compared to the full state are standard in related works (see Parisotto and Salakhutdinov). We have made them explicit in the Introduction and Preliminaries section in the updated version of our paper.
>
> Emilio Parisotto and Ruslan Salakhutdinov. Neural map: Structured memory for deep reinforcement learning.arXiv preprint arXiv:1702.08360, 2017.
>
> -- ”The reasoning related to mutual information seems circular.”
> The mathematics is used to derive the correct loss function that will maximize the mutual information as desired in the intuition in the beginning of the section (and provide proof that it actually does so).
>
> -- ”An intrinsic reward based on image reconstruction could drive the agent’s exploration during an initial pretraining phase, “
> We are unsure of what the suggestion here is. Is the image reconstruction reward being used to train the task RL agent or the glimpse agent?

---

### Official Review · AnonReviewer2 · 2020-10-28

**Rating:** 4
**Confidence:** 3

**Review:**

Summary

This paper proposes a new architecture and training method to learn tasks that require hard attention control. Specifically, the paper proposes to learn the “glimpse agent” (which controls the hard attention window) by task-agnostic loss that seeks to maximize information gain by the glimpse to the learned world model. The authors also proposed a specific architecture that incorporates consecutive glimpses to learn the world model.

==============

Positives

The paper is written clearly and cleanly, with helpful diagrams.

The mutual information objective the authors proposed to learn the glimpse agent is novel and intuitive.

==============

Negatives

Experiments are not thorough at all. Two similar tasks are chosen, with relatively low level of complexity. As a comparison, the experiments in SpatialNet, a work that inspired the authors, include a more diverse range of tasks including Atari tasks and 3D tasks (PhysShooter3D) and are generally more difficult.

The existing experiment results are not convincing. While results from PhysEnv show clear advantage of the proposed approach, the gridworld task does not. The authors can strength the result by having more complex tasks where the previous approach (labeled “environment”) would not work well.

It’s useful to see more analysis of the experiment. For example, it’d be interesting to track the mutual informative objective, which the glimpse agent optimizes, over the course of training. It’s also helpful to see if the learned glimpse agent + DMM can indeed be used for learning different tasks (with same physics environment), as the authors claimed to be task-agnostic.

==============

Recommendation

Overall I find the experiment results to be unconvincing and therefore recommend a reject.

---

> ### Author Response · Authors · 2020-11-25
> **Response R2**
>
> Thank you for your feedback.
>
> With regards to the comparison of environments with SpatialNet, we would like to point out that our problem domain is significantly harder due to the partial observability. Only 1/16th of the state is visible at any time to the agent in our version of the PhysEnv domain, hence long sequences (25 steps in our case) must be observed in order to learn models of motions of the various objects in the scene. In the SpatialNet work, the entire state was visible at every step, and hence learning was made significantly easier.
>
> As pointed out by R1, we have drawn attention to an unexplored research area of reinforcement learning with hard attention and presented a novel solution for it. We have derived a technically sound loss based on a well-motivated mutual information objective and we have demonstrated its effectiveness in two environments.
>
> Thank you for your suggestion for tracking the mutual information objective. We have added graphs and analysis for it in Appendix section A.3 (page 19) in the updated version of the paper.

---

### Official Review · AnonReviewer4 · 2020-10-28
**Interesting approach to learning a hard attention controller using curiosity; not yet clear it is useful on challenging tasks**

**Rating:** 4
**Confidence:** 4

**Review:**

This work presents a method for learning a hard attention controller using an information maximization approach. As the authors point out, such a method could be very useful for reasoning in terms of high-dimensional observations, like vision. In brief, the method learning to choose the next attention position to be the most informative by maximizing the uncertainty of the next observation. Uncertainty is quantified using a spatial memory model that is trained to reconstruct and predict the scene. The authors validate this approach by showing that the resulting attention mechanism can be used for two simple downstream tasks. The resulting agent outperforms others trained using baseline attention mechanisms: a hard attention mechanism that is trained on task reward ("environment"; similar to Mnih et al 2014), as well as models that attend to random positions or to the agent's location.

This work is well motivated, easy to read, and appears technically correct. The information maximization objective is a sensible way to learn an attention mechanism, and I found the exposition very easy to follow.

The main appeal of the method is that it's trained independently of the task, and for this reason might be useful on many tasks. And indeed, the authors highlight that this is the promise of this line of research (" ...our approach is unsupervised in terms of the task
and can be applied generally to downstream tasks in the environment"). Accordingly, my main concern is that the paper presents little evidence that the attention mechanism presented here will work in a task-agnostic manner. The paper only shows results on two simple 2D environments, and one of these evaluations has caveats that I feel significantly weaken the paper's case.

In particular, I don't find the comparison to baselines on the PhysEnv environment to be fair. This is for two reasons:
(1) while the strongest baseline method is trained on data from the learning agent, the proposed method is trained on data from an expert policy (as described in appendix section A) instead of from random transitions. In the typical RL setup, we can't generally assume that expert demonstrations will be available when training an agent from scratch, so this restricts the applicability of the attention mechanism for RL. The resulting attention policy is likely to indirectly leak task information to the agent, which makes it hard to compare to models trained without expert data on task reward alone.
(2) the authors report that the strongest baseline (called "environment"; Mnih et al 2014) doesn't perform well because the resulting policy is entropic. It's difficult to evaluate whether this behavior is due to problems with the baseline method or with the hyperparameter settings of the RL algorithm used to optimize it. For example, PPO uses a policy entropy bonus, and the authors don't report tuning this hyperparameter, but it will presumably play a large role in the entropy level of the learned attention mechanism. Because of this, it's hard to know whether the proposed methods outperforms the baseline because the baseline is poorly tuned or because the proposed method is generally better. I'm generally surprised by how poor the results of the "environment" method shown here are, given the it performs comparably on the other task and given that it's trained on task reward. More analysis or discussion would be very useful.

At a more fundamental level, I'm not convinced that the task-agnostic strategy for information maximization proposed here is the correct one for all tasks. This method will suffer from the "noisy TV" problem faced by many curiosity-based methods (as described e.g. in the introduction to Savinov et al 2019: https://arxiv.org/abs/1810.02274) and will attend to regions of high variability whether or not they're task relevant. In settings where the task of interest involves direct interaction with a relatively small number of pixels, but other things in the scene are also changing, there is no guarantee that an information maximization strategy will attend to the most task-relevant pixels. Without evaluation on more challenging tasks, it's hard to know how good a strategy information maximization is. The paper should address these issues explicitly. These results would be even stronger if shown on a perceptually harder task, such as one involving natural scenes, 3D content, or more realistic dynamics.

Other questions and comments (less central to my evaluation):
- What happens if the attention policy and the agent are trained simultaneously? If this approach works, it might allow the attention model to be trained on PhysEnv without requiring expert demonstrations.
- How does the "random" attention baseline perform on the two tasks? On PhysEnv, the random baseline gives the second best reconstruction results, so it would be very interesting to see how well it works for control.
- The claims made about human cognition in the introduction need to be better justified with references to the literature. The one reference provided (Barrouillet et al 2004) is primarily about working memory spans and cognitive load (i.e. internal bottlenecks) rather than about perceptual bottlenecks and the need to build world models or use hard attention mechanisms, as the surrounding text implies.
- The reconstruction results in section 6.1 are a good sanity check of the proposed model, but the baselines used here are very weak. This is because the proposed method uses an attention mechanism that's trained for reconstruction (via the infomax objective), while the other methods are trained either for an RL task (which may be only loosely correlated with reconstruction) or are heuristic. These results would be much more compelling with stronger baselines (and model ablations).
- Generally, the paper would benefit from more analysis of the contribution of model components. E.g. how important is the architecture of the dynamic memory module to the reconstruction and RL results?

Minor:
- Section 4.1: "This quantity is the amount of surprise in the dynamics model and we will use of this again when training the glimpse agent." -> "...we will use this again..."

- Section 4.2: "In addition to reconstruction loss," -> "In addition to the reconstruction loss,"

- Section 4.2: "The total loss for as single step" -> "The total loss for a single step"

In summary: the authors present an interesting application of information maximization-based curiosity to hard attention control. A hard attention mechanism trained in a purely unsupervised fashion (as proposed here) that performs well on many downstream tasks would be very useful. As it stands, I am in favor of rejecting this paper because of the limitations of the evaluation and analysis. My concerns would be addressed by evaluating on more challenging, benchmark tasks and ideally on tasks with more challenging visual structure, and with a thorough analysis of how the model performs in settings where information maximization is uncorrelated with the task (as in the "noisy TV" problem).

---

> ### Author Response · Authors · 2020-11-24
> **Response R4**
>
> Thank you for your particularly detailed review of our work. We have updated our submission thanks to your suggestions, with particulars outlined below.
>
> --”(1) while the strongest baseline method is trained on data from the learning agent, the proposed method is trained on data from an expert policy (as described in appendix section A)”
> All baselines (including ‘environment’) were trained on the same (expert) data. The expert’s actual policy was never used to train the eventual RL agent. An expert was used for data collection in PhysEnv solely to ensure that episodes do not end too quickly. Another data collection technique would be to rely on random transitions but discard all episodes below a certain length (25 in our case). To demonstrate that our method would still be able to learn in such a case we have added results from the “online” version of our method in figure 4 (left). In this version, the glimpse and memory are trained online along with the task solving RL agent, starting with random weights for all networks. In this version, glimpse and memory are trained on a separate replay buffer maintaining recent episodes longer than 25 steps. The online version of our method still learns a policy with reward approaching that of the pre-trained glimpse and memory version.
>
> -- “For example, PPO uses a policy entropy bonus, and the authors don't report tuning this hyperparameter,...”
> Both the "environment" baseline and our approach are trained using the same policy entropy bonus in PPO (0.001). Environment (or task) rewards for training glimpse work for small environments (as shown by Mnih et al. and our gridworld) but do not scale to complex inputs (more objects with complicated dynamics) because of the challenging credit assignment between completing the task and better perception of the environment. Please also refer to the qualitative comparisons between the baseline policies and our method added in the Appendix (figures 7 and 8).
>
> -- “This method will suffer from the "noisy TV" problem faced by many curiosity-based methods ..."
> We agree with the reviewer that all curiosity-based approaches tend to suffer from the “noisy TV” problem. Orthogonal approaches have been developed to counter this common weakness by, for example, taking into account the agent’s own actions (Pathak et al. 2017). These approaches can be combined with our method as well. We have proposed a novel (as noted by the reviewer) curiosity-based approach to attention control and leave incorporation of the optimizations from a complex approach to future work. We have updated our conclusion to make this clear.
>
> Pathak, Deepak, Pulkit Agrawal, Alexei A. Efros, and Trevor Darrell. "Curiosity-driven exploration by self-supervised prediction." In Proceedings of the IEEE Conference on Computer Vision and Pattern Recognition Workshops, pp. 16-17. 2017.
>
> --“What happens if the attention policy and the agent are trained simultaneously?”
> Thank you for the suggestion. We have incorporated this in the results (figure 4 left). Please also see the first discussion point above.
>
> -- ”How does the "random" attention baseline perform on the two tasks?”
> We are running experiments for this and will update the paper if they finish before the response period. For qualitative comparison, please look at figures 7 and 8 in the appendix which show that although the reconstruction error for “random” appears small, the full state reconstructions by our method are better (especially in PhysEnv). Hence the resulting memory representation for the “random” baseline is less accurate than our method’s, which is then used for performing RL on downstream tasks.
>
> -- ”The claims made about human cognition in the introduction need to be better justified…”
> Thank you for the suggestion. We have updated the citations for that section.
>
> -- ”The reconstruction results in section 6.1 are a good sanity check of the proposed model, but the baselines used here are very weak.”
> -- ”Generally, the paper would benefit from more analysis of the contribution of model components.”
> The “random” baseline is a model ablation and “environment” is a baseline from previous work (Mnih et al., 2014) . We have tried to incorporate the ablations and comparisons we can think of, if you could suggest further concrete comparisons we can attempt them.
>
> Here are preliminary results from running ablations on the memory architecture in PhysEnv.
> Method - Overall Reconstruction Error
> LSTM - 0.0885
> Spatial - 0.0562
> Ours - 0.0521
> Where LSTM corresponds to using a flat LSTM instead of a 2D structured memory and Spatial corresponds to the Spatialnet architecture without modifications for partially observable domains.
>
> Thank you for the minor suggestions!

---

### Author Response · Authors · 2020-11-24
**Overall Response**

We thank the reviewers for their detailed and helpful feedback. We have updated our submission following the suggestions by the reviewers (details provided in individual responses).

We also thank them for their encouraging words about the work, describing it as “well motivated”, “easy to read”, “technically correct”, “novel and intuitive”, “interesting and clear”, and “investigates the unexplored research area”. We are pleased to see our motivation and approach generally described as novel and correct and emphasize that this work is addressing a relatively overlooked area of reinforcement learning with a hard attention constraint.

---

### Decision · Program_Chairs · 2021-01-07
**Final Decision**

**Decision:**

Reject

**Comment:**

Inspired by biological agents that have developed mechanisms like attention as an information bottleneck to help function more effectively under various constraints of life, this paper looks at an approach of learning a hard attention scheme by leveraging off the prediction errors of an internal world model. They demonstrate their approach via a simple but easy to understand 2D pixel multi-agent game, a gridworld env, and also PhysEnv, to show the effectiveness of the learned hard attention, and go on to discuss interesting aspects such as curiosity attention.

Overall, I thought more highly of the paper than the reviewers, and might have proposed a score of 6 if I were a reviewer, but I also read each review and respect the points given by all four reviewers, and also agree with much of their feedback in the end. I think if this work was submitted to ALife (Journal or Conference), it might have been accepted. Not that those venues are easier, if anything they can often be more selective, but I think what ICLR (and similar venues like ICML) tends to expect is a bit different than what this paper offers.

To improve this work, I recommend following some of the reviewers' advice (especially R4), particularly on experimental design. Reviewers suggest that the current experiments are small and simple, but while true to some extent, I think more importantly missing are clear baseline methods to compare your approach against. What can your approach do that existing popular approaches in RL will totally fail at doing? It can be worthwhile to try your approach on a larger task domain, such as Atari (but perhaps modified) or ProcGen [1] to show the benefits of hard attention compared to existing approaches. For instance, some recent work [2] demonstrated that hard attention can help agents generalize to out-of-training domain tasks the agent has not seen before during training - something that traditional approaches without attention tend to fail at doing.

In the current state, the work will be a great workshop paper. But I recommend the authors to continue improving the work in the direction that can help the idea gain acceptance by the broader community.

[1] https://openai.com/blog/procgen-benchmark/
[2] https://arxiv.org/abs/2003.08165